# Protocol for analysing the epidemiology of maternal mortality in Zimbabwe: A civil registration and vital statistics trend study

Reuben Musarandega[1]☯*, Rhoderick Machekano[2‡], Robert Pattinson[3‡], Stephen Peter Munjanja[4☯], Zimbabwe Maternal and Perinatal Mortality Study (ZMPMS) group[¶]

1 School of Health Systems and Public Health, University of Pretoria, Pretoria, South Africa, 2 Biostatistics and Epidemiology Department, Faculty of Medicine and Health Sciences, Stellenbosch University, Cape Town, South Africa, 3 Research Centre for Maternal, Fetal, Newborn & Child Health Care Strategies, SAMRC Maternal and Infant Health Care Strategies Research Unit, University of Pretoria, Pretoria, South Africa, 4 Department of Obstetrics and Gynaecology, University of Zimbabwe College of Health Sciences, Harare, Zimbabwe

☯ These authors contributed equally to this work.
‡ These authors also contributed equally to this work.
¶ Membership of the Zimbabwe Maternal and Perinatal Study groups for 2007 and 2018 is listed in the Acknowledgments.
* rmusara@gmail.com

**Funding:** This work is supported, in part, by the Bill & Melinda Gates Foundation through a grant to the Improving Maternal Health Measurement (IMHM) Project at the Women & Health Initiative of the Harvard T.H. Chan School of Public Health [Grant Number OPP1169546]. It is also funded from the

## Abstract

### Background

Sub-Saharan Africa (SSA) carries the highest burden of maternal mortality, yet, the accurate maternal mortality ratios (MMR) are uncertain in most SSA countries. Measuring maternal mortality is challenging in this region, where civil registration and vital statistics (CRVS) systems are weak or non-existent. We describe a protocol designed to explore the use of CRVS to monitor maternal mortality in Zimbabwe—an SSA country.

### Methods

In this study, we will collect deliveries and maternal death data from CRVS (government death registration records) and health facilities for 2007–2008 and 2018–2019 to compare MMRs and causes of death. We will code the causes of death using classifications in the maternal mortality version of the 10th revision to the international classification of diseases. We will compare the proportions of maternal deaths attributed to different causes between the two study periods. We will also analyse missingness and misclassification of maternal deaths in CRVS to assess the validity of their use to measure maternal mortality in Zimbabwe.

### Discussion

This study will determine changes in MMR and causes of maternal mortality in Zimbabwe over a decade. It will show whether HIV, which was at its peak in 2007–2008, remains a significant cause of maternal deaths in Zimbabwe. The study will recommend measures to

UNDP-UNFPA-UNICEF-WHO-World Bank Special
Programme of Research, Development and
Research Training in Human Reproduction (HRP),
a cosponsored programme executed by the World
Health Organization (WHO). The Zimbabwe
Ministry of Health and Child Care is also
cosponsoring the study through funding from its
partners.

**Competing interests:** The authors have declared
that no competing interests exist.

improve the quality of CRVS data for future use to monitor maternal mortality in Zimbabwe
and other SSA countries of similar characteristics.

## Introduction

WHO defines maternal death as the death of a woman while pregnant, giving birth or within
42 days of termination of pregnancy, irrespective of the duration and site of the pregnancy,
from any cause related to or aggravated by the pregnancy or its management but not from
accidental or incidental causes [1]. WHO also defines the cause of death as all those diseases,
morbid conditions or injuries which either resulted in or contributed to death and the circum-
stances of the accident or violence which produced any such injuries [2]. Maternal deaths
remain a major global health problem [3–5]. In 2017, there were an estimated 295,000 mater-
nal deaths globally, 196,000 (66%) of which occurred in Sub-Saharan Africa (SSA), primarily
due to avoidable causes [3].

Sustainable Development Goal (SDG) 3.1 aims to reduce the average global maternal mor-
tality ratio (MMR) to less than 70 maternal deaths per 100,000 live births by 2030. Countries
with a 2010 baseline above 420 should not have an MMR greater than 140 maternal deaths per
100,000 live births [6, 7]. Attaining the SDG 3.1.1 global target by 2030 is a mammoth task for
SSA countries such as Zimbabwe, where the MMR is estimated above 400 maternal deaths per
100,000 live births [8–10]. More importantly, data will be required from all countries to mea-
sure progress towards SDG 3.1 [11], yet many countries lack comprehensive data systems to
measure maternal mortality [12].

Maternal mortality data come from civil registration and vital statistics (CRVS), routine
health information systems (HIS), population censuses, surveys and statistical modelling [13,
14]. SSA countries rely on modelling and population-based surveys that use the sisterhood
method for their MMR estimates, all of which are unreliable [14, 15]. Estimates that models
produce require countries to provide input data. These data are often unavailable in most SSA
countries [3, 16, 17]. Sisterhood studies collect data for deaths from the past seven or more
years; hence, they are prone to recall bias. Study respondents may not provide precise dates
and causes of death. Hence sisterhood studies only identify pregnancy-related deaths and not
maternal deaths. This makes sisterhood studies under-or over-estimate maternal mortality lev-
els [14, 16].

As with most SSA countries, Zimbabwe estimates MMR using periodic population-based
surveys that use the sisterhood method. These include Demographic and Health Surveys
(DHS), Multiple Indicator Cluster Surveys (MICS) and population census. The country has set
up a maternal death surveillance and response system (MDSR) following WHO recommenda-
tions, which is not fully functional [18].

CRVS data are widely used to estimate MMR in developed countries, but the first study
to use CRVS in Zimbabwe was done in 2007–2008. The study combined CRVS (data from
government death registration records) with data from health records (patient charts and ante-
natal care, delivery and mortuary registers) and the community (using verbal autopsy ques-
tionnaires) [19]. In Zimbabwe, CRVS is backed by legislation that mandates the notification,
registration by the government and issuance of a certificate for births and deaths of citizens
and residents.

In this study, we will analyse trends in MMR and causes of maternal deaths in Zimbabwe
using data for 2007–2008 and 2018–2019 to see if the country's MMR and causes of maternal

mortality have changed in the past decade. We will assess changes in the contribution of HIV to maternal mortality since 2007–2008, when HIV mortality was at its peak in the country. We will also evaluate the validity of using CRVS to monitor maternal mortality in Zimbabwe and recommend measures to ensure its future use in SSA countries of similar characteristics.

## Materials and methods

### Study aim

To analyse the epidemiology of maternal mortality in Zimbabwe by assessing changes in the MMR and causes of maternal death over a decade and the validity of using CRVS data in future to monitor maternal mortality in Zimbabwe and similar countries.

### Study objectives

1. To estimate the MMR for Zimbabwe in 2007–2008 and 2018–2019 and quantify the difference in MMR from these two time periods.

2. To assess changes in causes of maternal mortality in Zimbabwe between 2007–2008 and 2018–2019.

3. To assess changes in HIV-associated maternal mortality in Zimbabwe between 2007–2008 and 2018–2019.

4. To assess the validity of using CRVS to estimate national MMR and recommend measures to ensure their future use to monitor maternal mortality in Zimbabwe and similar countries.

### Study design and setting

We will conduct an observational, population-based, trend analysis study comparing maternal mortality estimates and cause for two studies that are ten years apart.

### Study methods

The study will use the reproductive age mortality survey (RAMOS) method, collecting data for all deaths of women of reproductive age (WRA) to identify maternal deaths [20, 21]. It will collect deaths for the one year from 1 May 2007 to 15 June 2008 and 1 May 2018 to 15 June 2019. The data will be collected from CRVS and health facilities (central, provincial, district, mission and rural hospitals and primary care clinics).

### Study population

Women of reproductive age, 12–49 years in Zimbabwe.

### Study outcomes

The primary study outcomes are change in MMR and changes in proportions of deaths due to the leading causes of maternal mortality in Zimbabwe from 2007–2008 to 2018–2019. Secondary study outcomes are changes in proportions of deaths of WRA and pregnancy-related deaths.

### Sampling methods and sample size calculations

The sampling for 2007–2008 and 2018–2019 data is the same. The sample size was estimated using deliveries—the denominator for the MMR. The population was stratified into the ten

provinces of the country, and one district was selected from each province using simple random sampling [22–24]. An extra district was sampled from Harare by proportional-to-size sampling, considering the city's large population and two referral hospitals. In total, the study will be done in 11 districts and 291 health facilities.

The sample size of deliveries was estimated using the last verifiable MMR of Zimbabwe and the country's total fertility rate (TFR), treating MMR as a single proportion [25]. It was calculated using the Wald test, which estimates a one-sample proportion against an expected proportion (the last verifiable MMR as a proportion). Power of 80%, z-value for two-sided 95% CI, normal-approximation continuity correction for the expected proportion, 2.5% error margin (for the alternative hypothesis of MMR outside the 95% CI of the last verified MMR) was applied. A design effect of 2 was also applied, based on the two-step sampling procedure of stratifying the study population into provinces and districts and randomly selecting the districts [23, 24, 26]. Deliveries and deaths occurring in the study districts are consecutively enrolled until the sample size of deliveries is reached.

## Data collection procedures

The data collection procedures are summarised in Fig 1.

**Procedures for 2007–2008 data collection.** The 2007–2008 dataset will be built from data collected in the first study conducted in 2007–2008. We will validate the 2007–2008 data by revisiting the vital registration (VR) and hospital records in the 11 districts and collect data for all deaths of WRAs and their causes of death for 2007–2008. We will collect case notes for the deaths from hospitals to code the causes of death. We will use the new data to clean up the data collected in 2007–2008.

**Procedures for 2018–2019 data collection.** We will collect the 2018–2019 data from the same 11 districts using the same method recording all deaths of WRA at the VR offices and hospitals. Depending on the level of the institution, sources of maternal deaths data in health institutions in Zimbabwe include maternal death line-list in reproductive health offices, delivery unit, casualty unit, theatre, intensive care unit, high dependency unit, female ward, and mortuary registers [27]. We will triangulate and collect additional data from the maternal death surveillance, and response (MDSR) reports. MDSR reports and VR records include community deaths. Deaths of WRA will be identified at the VR offices from birth and death registration forms filed by year in locked records rooms. We will record the data on line-listing forms, including demographic and cause of death information for each woman. A second line-listing form will record more detailed data for each PRD, including personal identifiers such as name, date of birth, national identification number, home address, place of residence, date of death, place of death and cause of death. We will use personal identifiers to track and de-duplicate the women across CRVS and health facility records.

PRDs will be identified at health facilities from maternity, delivery, maternal death and mortuary registers, and MDSR reports. The data will be recorded on the same line-listing forms. Case notes for all PRDs will be collected from hospitals. MDSR reports will be collected from the district and provincial health offices. PRDs identified in the RAMOS will be verified with those reported in the district health information system (DHIS2) by reporting month and institution to avoid missing some deaths. The 2007–2008 data were collected on a form adapted from WHO. We will use updated versions of the 2007–2008 questionnaire to collect the 2018–2019 data and use the newly collected 2007–2008 data to update the previous data forms. Variables collected in the form include demographic, antenatal, antepartum and postpartum information and causes of death. Four (4) trained research midwives will collect the data from all districts.

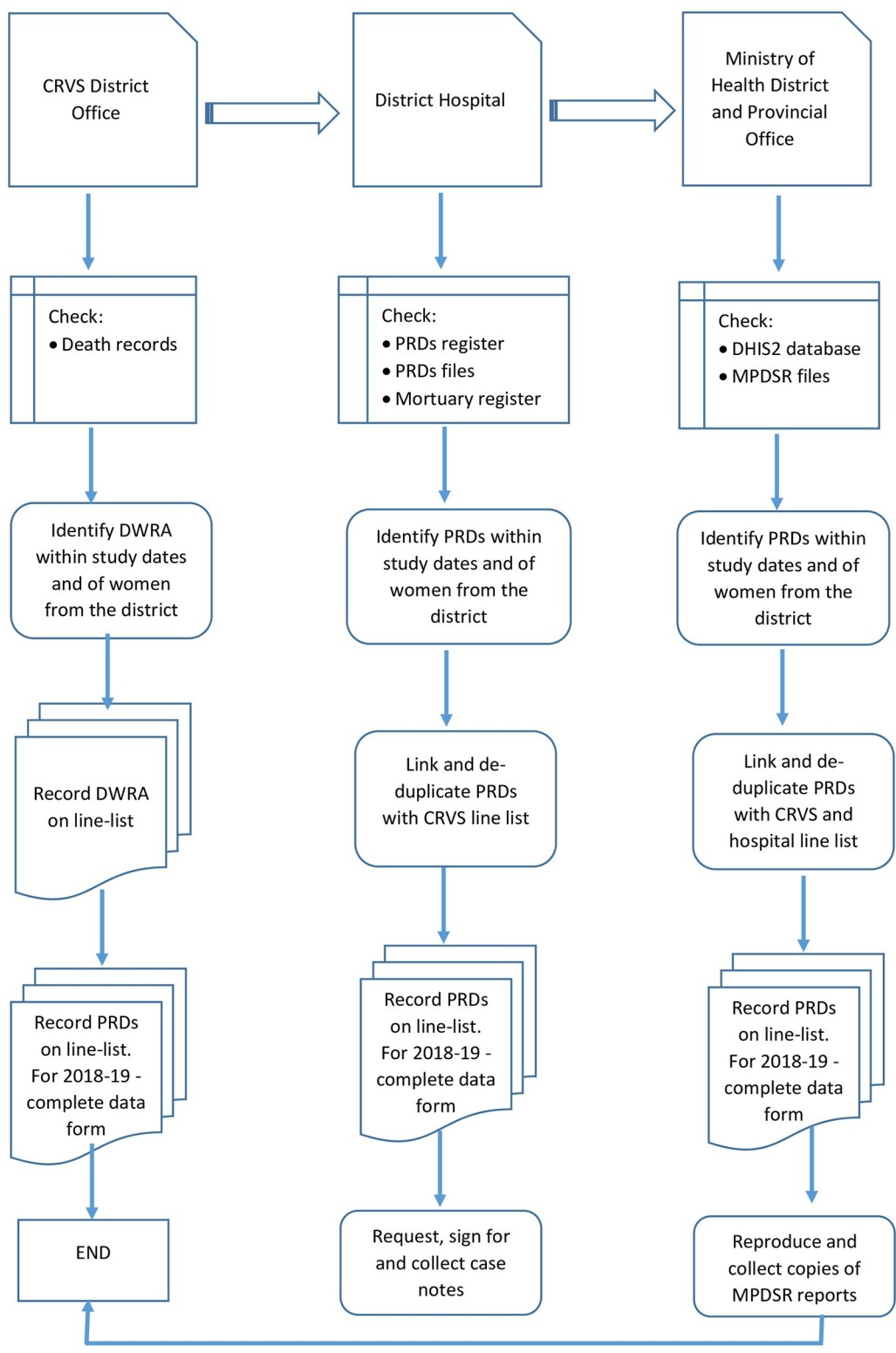

**Fig 1. Data collection procedures for 2007–2008 and 2018–2019.**

**Coding of the causes of death in 2007–2008 and 2018–2019 pregnancy-related deaths (PRDs).** A panel of obstetricians will review all PRDs for 2007–2008 and 2018–2019 to confirm maternal deaths, the type of death (direct, indirect or incidental), category and the actual cause of death. They will use the WHO ICD-10 MM guide for the classification and coding. ICD-10 MM groups the causes of maternal death into nine groups, namely: 1: Pregnancies with the abortive outcome, 2: Hypertensive disorders in pregnancy, childbirth and the puerperium, 3: Obstetric haemorrhage, 4: Pregnancy-related infection, 5: Other obstetric complications, 6: Unanticipated complications of management, 7: Non-obstetric complications, 8: Unknown/undetermined causes and 9: Coincidental causes [1]. Groups 1–8 are maternal deaths, while group 9 are pregnancy-related deaths. Obstetrician reviews will be documented on an additional data form page attached to each PRD's data form. The primary data form and line-listing forms were adapted from the WHO maternal mortality and morbidity systematic review tool [28]. The death registration forms used in Zimbabwe are aligned to ICD-10 MM.

## Data management

For 2007–2008 data, the additional data form page will be used to facilitate changes in the database. For 2018–2019, the line-listings of PRDs will be used to complete the main data form. The data form is completed in the field while data collectors still have access to other patient data from field records. After coding the causes of death, the data will be entered into a password protected MS Access database. The data processing procedures are summarised in Fig 2.

## Inclusion and exclusion criteria

The study will include deaths of WRA (12–49 years) from the study districts, including those who died during pregnancy or within 100 days of termination of pregnancy. We will include PRDs due to illnesses, injuries and accidents that were not pregnancy-induced. We will exclude these from MMR estimates. We will analyse these other deaths of WRA as secondary outcomes. Maternal deaths of unknown or non-specific causes coded in ICD-10 MM group 8 will be included in MMR estimates and cause of death analyses. Women who died within but originated from other districts will be excluded from MMR estimates and cause of death analyses but reported in secondary outcomes.

## Quality control measures

The principal investigator (PI) will collect data with the research midwives at the first two districts and at two other districts midway through data collection. During this time, the PI will ensure that data collectors are following the protocol. PRDs identified in the study will be linked across all data sources to minimise missing data and avoid duplicating some deaths. The PI will review the completeness and quality of data on all forms. Obstetricians' review of every PRD will be another quality control measure on the data.

## Data analysis

**Descriptive analysis.** We will compare demographic and clinical characteristics of 2007–2008 and 2018–2019 deaths, including age, parity, gravidity and pregnancy complications, using proportions and mean/medians (with standard deviations) with tests of significance of difference.

**Multivariable logistic regression.** We will assess risk factors for maternal death using multivariable logistic regression models on 2007–2008 data, which has a comparison group of deliveries where the women did not die. We will include predictor variables with univariate p-values less than 0.25 in the initial model and test model fitness using Hosmer and Lemeshow

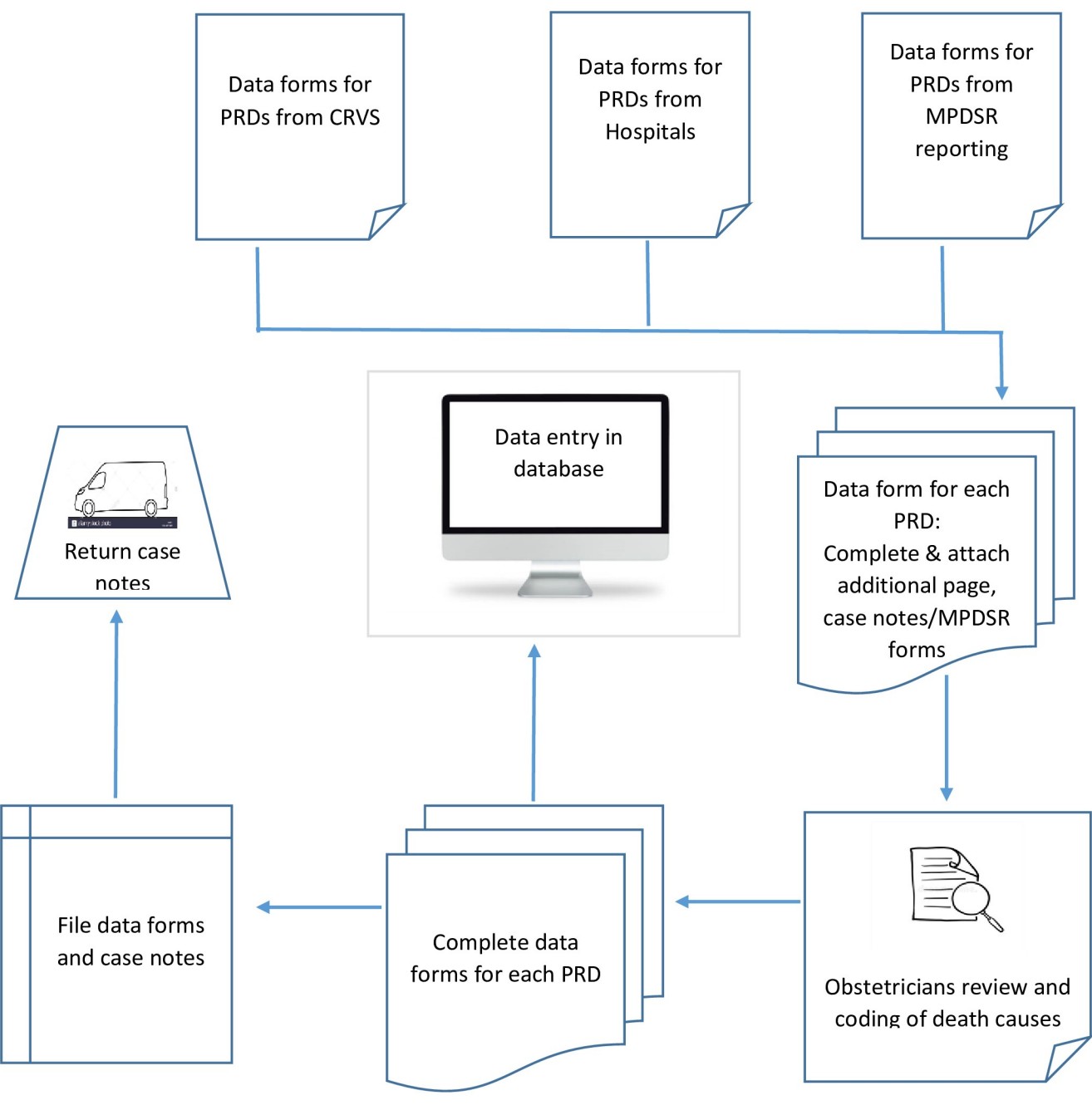

**Fig 2. Data processing procedures.**

Chi-square test [29]. Predictor variables with p-values less than 0.2 will be included in the final model [30]. This analysis will be design-adjusted and use survey weights [31, 32].

**Changes in MMR.** We will calculate and compare MMRs for 2007–2008 and 2018–2019 using their 95% CIs. We will calculate the MMRs by dividing the total number of maternal deaths with the total live births for that year, multiplied by 100,000 [33, 34]. MMR estimates with overlapping CIs will suggest that the MMR has not changed over the decade. However, we will also consider the clinical significance of differences in MMRs when CIs show no

difference. Differences in MMR point estimates greater than 100 will be considered clinically significant [35, 36].

**Changes in causes of death.**    We will calculate proportions (with 95% CIs) of deaths due to each underlying cause of death by dividing the number of deaths for each cause with the total number of deaths in the survey to identify the leading causes of maternal mortality in 2007–2008 and 2018–2019. We will use the 95% CIs to evaluate changes in the contribution of each cause to maternal mortality over the decade [37]. Subject to the suitability of the study design and availability of requisite data, competing risk analysis will be considered in cause-of-death analysis [38].

**Changes in HIV-associated mortality.**    We will calculate and compare the proportion of HIV-associated maternal deaths in 2007–2008 and 2018–2019. HIV-associated maternal deaths are deaths where an AIDS-defining condition is present [28, 39, 40]. We will calculate the proportions by dividing the number of HIV-associated maternal deaths by the total number of maternal deaths in that year.

**Validity of using CRVS to monitor maternal mortality and recommendations for future use.**    We will triangulate the number of maternal deaths identified from CRVS, health facilities and MDSR to quantify missingness in the study. We will classify the maternal deaths according to false positives, false negatives, true positives, true negatives, and missing (both true and false maternal deaths) to assess the validity of using CRVS to monitor maternal mortality in Zimbabwe. We will review the quality of records in CRVS records to evaluate its use to monitor maternal mortality. We will discuss the challenges identified in the data systems and recommend measures to strengthen CRVS for future use to monitor maternal mortality in Zimbabwe and similar countries.

**Weighting the survey data.**    Some of our analysis will be weighted since this study has multistage sampling [41–43]. The weights will be calculated using non-proportional sample allocation to clusters and possible inter-cluster differences in response rates [22, 44]. The weights are calculated using the product of two probabilities, considering the two clustering levels of province and district in the sampling design. The first probability is for selecting a district in a province, which is one out of the number of districts in the province. The second probability is for identifying a maternal death in the district, estimated at 70% (0.7) by a verification exercise conducted in the 2007–2008 study [19]. The final sample weight is the inverse of the product of the two probabilities [44, 45]. Sample weights will be calculated per district since the probabilities vary by district.

**Ethics approvals.**    The protocol was approved by the UNDP-UNFPA-UNICEF-WHO--World Bank Human Reproduction Program (HRP) (Date: 2019-03-27), World Health Organization Ethics Review Committee (Ref: ERC 0003348; Date: 2020-04-24), Medical Research Council of Zimbabwe (Ref: MRCZ/A/2613; Date: 2020-07-17) and University of Pretoria Faculty Health Research Ethics Committee (Ref: 339/2019; First submission approved– 2020-07-15; Amendment approved: 2021-01-21). The Ministry of Health and Child Care (MoHCC) and the Registrar General's Department of Zimbabwe granted permissions for access to patient and civil registration records, respectively. Additional clearances for data collection will be received from provincial, district and station heads of the MoHCC and RG's offices upon presentation of permission letters from their head offices. All approvals waived consenting of study participants since the study will collect data from existing records and not engage living human subjects.

## Discussion

This study will assess the changes in MMR and causes of maternal deaths over a decade in a developing country. The study will assess changes in the contribution of HIV to maternal

mortality since 2007–2008 when HIV mortality was at the peak in the country. The study will assess the completeness and accuracy of maternal death and cause-of-death data collected from different sources to explore the validity of using CRVS to monitor maternal mortality in Zimbabwe.

Our literature review indicated that few countries had estimated MMR using RAMOS methods and CRVS data in SSA [46]. Adomako in Ghana and Mgawadere in Malawi used the RAMOS method to estimate MMR for a district [47, 48]. Mohammed in Sudan, Walraven in Gambia and Zakariah and Geynisman in Ghana used the method to estimate MMR in urban and rural communities [49–52]. These studies did not use CRVS data as the systems did not exist in the countries. This will be one of the few studies to use the RAMOS method with CRVS data to produce a national MMR estimate.

Evaluation of decadal changes in MMR and causes of maternal mortality has not been done in many SSA countries. Other studies have shown that haemorrhage, hypertensive disease in pregnancy, pregnancies with abortive outcomes and pregnancy-related infections are the common causes of maternal mortality in SSA [53]. The contribution of caesarean sections to maternal mortality has been reported in Zimbabwe and other SSA countries like Sierra Leone [54, 55]. The contribution of epidemics like Ebola to maternal mortality has been observed in West African countries. Zimbabwe had a similar epidemic of Cholera in 2009–2009, which may have contributed to maternal mortality [56]. Preliminary results from the 2007–2008 survey suggest that HIV was the major cause of maternal mortality at that time [19]. HIV prevalence was high, and HIV treatment was still in the early rollout phase during these years [57]. However, Zimbabwe has since rolled out antiretroviral therapy (ART) to most of its health facilities, and ART coverage increased to more than 80% of people living with HIV [58]. Prevention of mother-to-child transmission (PMTCT) regimens was rolled out to nearly all hospitals and primary care clinics offering maternal and child health services in 2011–2012 and 2013, respectively [59]. The impact of ART on mortality has been observed in Zimbabwe [60], and elsewhere [61]. Thus, we expect ART to have reduced maternal mortality due to HIV, as seen in other SSA studies [62]. Other interventions implemented during this period, such as massive donor funding for maternal and child health programs through the health transition and health development funds, will be examined [63].

Our study will assess the feasibility of using CRVS to generate national MMR estimates. The study will recommend measures to strengthen CRVS for future use to produce national MMR estimates. The use of CRVS to monitor maternal mortality will strengthen the systems in the long run. It will make MMR estimates available annually, enabling the country to monitor trends. Our study will provide lessons for other SSA countries with similar characteristics as Zimbabwe.

## Acknowledgments

The authors acknowledge the Zimbabwe Maternal and Perinatal Mortality Study (ZMPMS) group members for 2007–2008 and 2018–2019 (Thulani Magwali, Margaret Nyandoro[†], Lennarth Nystrom, Bernard Madzima, Davidzoyashe Makosa, Chipo Chimamise, Michael Nyakura, Sunhurai Mukwambo). The support received from Jennifer Cresswell of WHO-Geneva and Jolivet Rima of the Human Reproduction Program in the study project and reviewing this manuscript is acknowledged. Assistance received from Cheryl Tosh of the University of Pretoria to copy-edit the article is also appreciated.

## Author Contributions

**Conceptualization:** Reuben Musarandega.

**Data curation:** Reuben Musarandega.

**Funding acquisition:** Stephen Peter Munjanja.

**Methodology:** Reuben Musarandega, Robert Pattinson, Stephen Peter Munjanja.

**Supervision:** Robert Pattinson, Stephen Peter Munjanja.

**Validation:** Rhoderick Machekano.

**Writing – original draft:** Reuben Musarandega.

**Writing – review & editing:** Rhoderick Machekano, Robert Pattinson, Stephen Peter Munjanja.

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
