## [Decision Letter · Decision Letter 0]

19 Apr 2021

PONE-D-21-07325

Protocol for analysing the epidemiology of maternal mortality in Zimbabwe: a civil registration and vital statistics trend study

PLOS ONE

Dear Dr. Reuben,

Thank you for submitting your manuscript to PLOS ONE. After careful consideration, we feel that it has merit but does not fully meet PLOS ONE’s publication criteria as it currently stands. Therefore, we invite you to submit a revised version of the manuscript that addresses the points raised during the review process.

We look forward to receiving your revised manuscript.

Kind regards,

Claudia Marotta

Academic Editor

PLOS ONE

Journal Requirements:

3. A reviewer has recommended that you cite specific previously published works. Please review and evaluate the requested works to determine whether they are relevant and should be cited. It is not a requirement to cite these works.

Additional Editor Comments:

dear authors follow reviewer suggestions to improve your already good paper

Reviewers' comments:

Reviewer's Responses to Questions

**Comments to the Author**

1. Does the manuscript provide a valid rationale for the proposed study, with clearly identified and justified research questions?

Reviewer #1: Yes

Reviewer #2: Yes

2. Is the protocol technically sound and planned in a manner that will lead to a meaningful outcome and allow testing the stated hypotheses?

Reviewer #1: Yes

Reviewer #2: Yes

3. Is the methodology feasible and described in sufficient detail to allow the work to be replicable?

Reviewer #1: Yes

Reviewer #2: Yes

4. Have the authors described where all data underlying the findings will be made available when the study is complete?

Reviewer #1: Yes

Reviewer #2: Yes

5. Is the manuscript presented in an intelligible fashion and written in standard English?

Reviewer #1: Yes

Reviewer #2: Yes

6. Review Comments to the Author

You may also provide optional suggestions and comments to authors that they might find helpful in planning their study.

Reviewer #1: I read with great interest this paper fromm Zimbawe. Authros wrote in my opinion an important paper. I worked in Sierra Leone on a project of maternal infection after caesarean section and all the research aorund maternal mortality are precious to improve maternal health and indicators.

Below my suggestion

1. introduction: add the role of infection after es caesaren infection in africa represent high burden for maternal mortlaity and disability (see and cite aternal caesarean section infection (MACSI) in Sierra Leone: a case-control study. Epidemiol Infect. 2020 Feb 27;148:e40. doi: 10.1017/S0950268820000370)

2. Method : very interesting

3. Discussion: discuss better also the role of HIgh dependenc unit and the need in africa of area where critial patients can be cured (see and cite Epidemiology, Outcomes, and Risk Factors for Mortality in Critically Ill Women Admitted to an Obstetric High-Dependency Unit in Sierra Leone. Am J Trop Med Hyg. 2020 Nov;103(5):2142-2148 ) and how obstestric critical care are also economic for health system with help for ex of NGO o others actors in health cooperation see and cite (Cost-Utility of Intermediate Obstetric Critical Care in a Resource-Limited Setting: A Value-Based Analysis. Ann Glob Health. 2020 Jul 20;86(1):82. doi: 10.5334/aogh.2907)

Furthermore, discuss how for example during pandemic (ebola or other epidemic , see Impact of Ebola outbreak on reproductive health services in a rural district of Sierra Leone: a prospective observational study. BMJ Open. 2019 Sep 4;9(9):e029093. doi: 10.1136/bmjopen-2019-029093. ) women are always most vulnerable group for all of these reason this paper is imporant

Reviewer #2: Overall, this protocol is well designed and complete.

Similarly, the programmed analysis in sound and well described.

Few minor revisions and questions:

1) Did some interventions were made to reduce maternal mortality in Zimbabwe between the two study period? It would be interesting to declare these interventions and perform a pre – post analysis

2) Please include Figure 1 and Figure 2 legend

3) Could be interesting to perform a competing risk analysis of death more than a “simple” logistic regression?

7. PLOS authors have the option to publish the peer review history of their article (what does this mean?). If published, this will include your full peer review and any attached files.

Reviewer #1: **Yes: **Francesco Di Gennaro

Reviewer #2: No

---

## [Author Response · Author response to Decision Letter 0]

21 Apr 2021

Response to Reviewers

Journal Requirements:

Comment 1: Please ensure that your manuscript meets PLOS ONE’s style requirements, including those for file naming.

Response: We have updated the author information (lines 3-4 and 14-20), acknowledgements (lines 321-323), section headings of the paper (Abstract, Introduction, Methods and Materials and Discussion), citation of figures (Fig 1 & Fig 2) and naming of files according to journal requirements (Figures, Revised manuscript with tracked changes, revised manuscript clean version and responses to reviewer comments). 

Comment 2: Please review your reference list to ensure that it is complete and correct.

Response: We updated some references with missing information. We did not identify any retracted references. If there is any, we will exclude it from our references. 

Reviewer #1 Comments

Comment 1: Introduction: add the role of infection after es caesarean infection in Africa represent a high burden for maternal mortality and disability (see and cite maternal caesarean section infection (MACSI) in Sierra Leone: a case-control study. Epidemiol Infect. 2020 Feb 27;148:e40. doi: 10.1017/S0950268820000370)

Response: We have adopted this recommendation and added the relevant information in the discussion, paragraph 3, lines 284-287. To place this suggestion in full context, we added other causes identified in the literature. 

Comment 2: Methods: very interesting

Response: We appreciated the comment and pleased that the study will contribute to maternal mortality research methodology. 

Comment 3: Discussion: Discuss the role of the high dependency unit and the need in Africa of an area where critical patients can be cured (see and cite Epidemiology, Outcomes, and Risk Factors for Mortality in Critically Ill Women Admitted to an Obstetric High-Dependency Unit in Sierra Leone. Am J Trop Med Hyg. 2020 Nov;103(5):2142-2148 ) and how critical obstetric care are also economical for a health system with the help forex of NGO and other actors in health cooperation see and cite (Cost-Utility of Intermediate Obstetric Critical Care in a Resource-Limited Setting: A Value-Based Analysis. Ann Glob Health. 2020 Jul 20;86(1):82. doi: 10.5334/aogh.2907)

Response: We adopted this comment and accepted the first reference. However, we thought it fits better in the methods. Therefore, we added it in the section where we mentioned the source records for data from health institutions, in the methods section, lines 148-151. 

Comment 4: Furthermore, discuss how, for example, during pandemic (ebola or other epidemics, see Impact of Ebola outbreak on reproductive health services in a rural district of Sierra Leone: a prospective observational study. BMJ Open. 2019 Sep 4;9(9):e029093. doi: 10.1136/bmjopen-2019-029093. ) women are always the most vulnerable group for all of these reasons; this paper is important.

Response: We adopted the comment and incorporated the suggested information in the discussion section, lines 288-290. Ebola has not affected Zimbabwe and Southern Africa at large. However, there has been a Cholera epidemic in Zimbabwe of similar magnitude and impact to Ebola. Hence we have added information on the Cholera epidemic in this section. 

Reviewer #2 Comments

Overall comment: Overall, this protocol is well designed and complete.

Similarly, the programmed analysis is sound and well described.

Response: We appreciated the comment 

Few minor revisions and questions:

Comment 1: We some interventions made to reduce maternal mortality in Zimbabwe between the two study period? It would be interesting to declare these interventions and perform a pre-post analysis

Response: We appreciated this comment and incorporated the suggestion. HIV interventions were already mentioned in the paper. However, we added funding for maternal and child health through two major multi-donor funds. We added this information in the discussion section, lines 299-301. 

Comment 2: Please include Figure 1 and Figure 2 legend

Response: We have added the legends to the figures. 

Comment 3: Could it be interesting to perform a competing risk analysis of death more than a “simple” logistic regression?

Response: We appreciated this suggestion and incorporated it in the data analysis section, lines 231-232. However, since competing risk analysis is a type of time-to-event or survival analysis, we are unsure if our study design and data will fit this type of analysis. We will explore it in study implementation. 

END OF COMMENTS

---

## [Decision Letter · Decision Letter 1]

10 May 2021

Protocol for analysing the epidemiology of maternal mortality in Zimbabwe: a civil registration and vital statistics trend study

PONE-D-21-07325R1

Dear Dr. Reuben,

We’re pleased to inform you that your manuscript has been judged scientifically suitable for publication and will be formally accepted for publication once it meets all outstanding technical requirements.

Kind regards,

Claudia Marotta

Academic Editor

PLOS ONE

Additional Editor Comments (optional):

congratulations for great paper

Reviewers' comments:

Reviewer's Responses to Questions

**Comments to the Author**

1. Does the manuscript provide a valid rationale for the proposed study, with clearly identified and justified research questions?

Reviewer #1: Yes

Reviewer #2: Yes

2. Is the protocol technically sound and planned in a manner that will lead to a meaningful outcome and allow testing the stated hypotheses?

Reviewer #1: Yes

Reviewer #2: Yes

3. Is the methodology feasible and described in sufficient detail to allow the work to be replicable?

Reviewer #1: Yes

Reviewer #2: Yes

4. Have the authors described where all data underlying the findings will be made available when the study is complete?

Reviewer #1: Yes

Reviewer #2: Yes

5. Is the manuscript presented in an intelligible fashion and written in standard English?

Reviewer #1: Yes

Reviewer #2: Yes

6. Review Comments to the Author

You may also provide optional suggestions and comments to authors that they might find helpful in planning their study.

Reviewer #1: Congratulations for your great paper. The setting and the research idea are high quality.

I appreciate a lot this new version of paper

Reviewer #2: No issues should be address before this protocol can be accepted for publication.

I have no further comments to made

7. PLOS authors have the option to publish the peer review history of their article (what does this mean?). If published, this will include your full peer review and any attached files.

Reviewer #1: No

Reviewer #2: No

---

## [Editor Report · Acceptance letter]

14 May 2021

PONE-D-21-07325R1 

Protocol for analysing the epidemiology of maternal mortality in Zimbabwe: a civil registration and vital statistics trend study 

Dear Dr. Musarandega:

I'm pleased to inform you that your manuscript has been deemed suitable for publication in PLOS ONE. Congratulations! Your manuscript is now with our production department. 

Kind regards, 

on behalf of

Dr. Claudia Marotta 

%CORR_ED_EDITOR_ROLE%

PLOS ONE